# Dynamical Task Switching in Cellular Computers

**DOI:** 10.3390/life9010014

**Published:** 2019-01-26

**Authors:** Angel Goñi-Moreno, Fernando de la Cruz, Alfonso Rodríguez-Patón, Martyn Amos

**Affiliations:** 1School of Computing, Newcastle University, Newcastle Upon Tyne NE4 5TG, UK; 2Instituto de Biomedicina y Biotecnología de Cantabria, Universidad de Cantabria, 39011 Santander, Spain; fernando.cruz@unican.es; 3Departamento de Inteligencia Artificial, Universidad Politécnica de Madrid, 28660 Madrid, Spain; arpaton@fi.upm.es; 4Department of Computer and Information Sciences, Northumbria University, Newcastle upon Tyne NE1 8ST, UK; martyn.amos@northumbria.ac.uk

**Keywords:** synthetic biology, cellular computing, plasmids

## Abstract

We present a scheme for implementing a version of task switching in engineered bacteria, based on the manipulation of plasmid copy numbers. Our method allows for the embedding of multiple computations in a cellular population, whilst minimising resource usage inefficiency. We describe the results of computational simulations of our model, and discuss the potential for future work in this area.

## 1. Introduction

Synthetic biology [1,2] is often broadly defined as the rational engineering of biological systems, with the aim of implementing novel computational functions in living organisms. Cells, such as bacteria, may host engineered networks of regulatory proteins—so-called genetic circuits—that sense inputs, perform processing, and generate outputs according to human-defined rules.

These artificial cellular computers are often (although not exclusively) single-purpose, in that they perform one well-defined task, such as the production of drug precursors [3] or the detection of environmental pollutants [4]. Interestingly, the representation of information by physical properties of a biological system (such as levels of gene expression) immediately suggests a parallel with analogue computers [5,6], which also used physical quantities, such as hydraulic pressure or the elasticity of a spring, to model specific problems. Importantly, analogue computers were generally geared towards specific applications, such as calculating bomb trajectories, and this did not limit their usefulness. That said, recent work in synthetic biology has acknowledged the need for more general-purpose biological computing platforms [7], capable of executing a diverse range of “algorithms” without significant modification.

Here, we are specifically interested in the possibility of engineering biological systems that are capable of task switching; that is, moving between a number of pre-programmed behaviours (or “tasks”), according to specific rules or signals. This allows for the possibility of embedding different tasks into a cellular population, while essentially performing dynamical resource allocation to ensure that the cells do not become metabolically over-burdened. We might compare this to the computer memory management strategy of “paging”, whereby inactive processes are moved from the (limited) main memory into secondary storage (such as a disk). In our model, “active” processes are represented by plasmids which exist in high numbers, and plasmids that are relatively few in number are considered to be “inactive”.

Here, we present a system in which multiple genetic circuits coexist, and control strategies select which one is functional (i.e., which task runs) at a given time. Our method is based on controlling the replication of plasmids, which are small DNA molecules which exist (and may be replicated) inside the cell independently of the main chromosomal material.

The rest of this paper is organized as follows: In Section 2, we provide some background to and motivation for our proposed method. In Section 3, we then present our main modelling and simulation results, describing our methodology in Section 4. We then conclude, in Section 5, with a discussion of our findings, and propose future work.

## 2. Background

Synthetic biology is a rapidly-growing scientific area, with applications in many significant domains, including health, energy, and the environment [8]. One branch of synthetic biology, which we call “cellular computing” [9], is specifically concerned with the construction of computational parts and devices using living cells [10,11]. These implementations include Boolean logic gates [7,12], switches [13], oscillators [14], and counters [15].

A fundamental tool in genetic engineering (and, thus, synthetic biology) are plasmids; these are small, circular DNA molecules used to introduce new genetic material into bacteria and other cells [16,17,18]. Typically, new genetic circuits are encoded as a number of genes, the sequences of which are then synthesised and inserted into the plasmid. Plasmids may also naturally be moved between cells via conjugation, facilitating a process known as Horizontal Gene Transfer (HGT) [19,20].

Horizontal gene transfer via conjugation has previously been proposed as a useful mechanism for performing computations [21], by using plasmids to transmit signals between cells. Here, we use the information processing capability of circuits, embedded in single or multiple plasmids [7], rather than simply using the plasmids to carry signals. This allows us to potentially run a number of different “programs” within the cell population, whilst ensuring that only active processes consume scarce system resources. This is the most important aspect of our proposal; while it is, in principle, possible to engineer multiple functional circuits into bacteria (and switch between them), in practice this is difficult to achieve, and inactive circuits place a significant metabolic overhead on the hosts. By dynamically switching between active circuits, and having only active circuits present in the host bacteria in significant numbers, we allow for flexible computational behaviour, whilst minimising the burden on the hosts. In the next section, we describe our model in detail.

### Our Task Switching Model

In previous work, we showed how individual cells may be engineered to exhibit different computational behaviours, according to the type of input received [22]. This essentially “flipped” a single genetic circuit between Boolean NAND and NOR, depending on input thresholds. Here, we maintain multiple circuits within a population of cells, and (de)activate them according to need.

A high-level description of our model is shown in Figure 1A. We have two basic levels of control; the lowest level concerns individual genetic circuits encoded in plasmids, and the higher (control) level switches between these circuits by manipulating the population dynamics of the plasmid pool. We focus mainly on this control level, as the embedding of computational circuits in plasmids is widely-studied [7].

Plasmid replication occurs using the host cell’s DNA replication machinery; plasmids contain sequences known as origins of replication (ori), which “instruct” the host cell to initiate its own replication. Importantly, a plasmid ori sequence may be activated or deactivated by initiator/repressor proteins which bind to it, thus turning on or off the replication of that plasmid [23]. In turn, initiator/repressor proteins may be expressed by genes in other plasmids, allowing one plasmid to effectively turn another on or off. Plasmids propagate in a system through either horizontal transfer by cell-cell conjugation, or vertically, when a cell divides into two daughter cells (Figure 2A. Plasmids that are being repressed are not transmitted vertically, but may still be transferred horizontally between siblings. Two specific attributes of plasmids are of direct interest: their copy number, and their stability. Copy number refers to the expected number of instances of a specific plasmid within a single host cell, and this may be “low” (15–20 copies per cell), “medium” (20–100 copies per cell), or “high” (>500 copies per cell). Engineered plasmids may be “set” to any preselected copy number, but there is an attendant trade-off: The higher the copy number, the higher the metabolic burden on the cell. Plasmid stability [24] exists when, at cell division, each daughter cell receives at least one copy of the plasmid.

In the next section, we describe the results of modelling and simulation experiments to investigate the properties and behaviour of systems constructed within our scheme.

## 3. Results

### 3.1. Continuous Modelling

In Figure 1B, we show the behaviour of a two-plasmid system, modelled using ordinary differential equations, in which plasmid A represses (i.e., “turns off”) the replication of plasmid B, which, in turn, replicates via positive feedback. We note a fragile equilibrium for the stability of plasmid B; there is only one scenario, $k_{1}/k_{2}$ = number of A (100) where the copy number of B does not either increase indefinitely or decrease to zero. This highlights the need for a stabilisation mechanism in such systems, which we describe below.

### 3.2. Discrete Simulation

In Figure 2, we show the results of discrete simulation of a population of cells containing two plasmids, A and B. Each plasmid’s computational “task” is not specified, since we focus here on the dynamics of copy numbers over time (red for plasmid A, and green for plasmid B). We emphasise the fundamental principle that plasmids that are repressed are not spread vertically (through cell division), but may still propagate horizontally [25] (Figure 2A). The significance of this is shown in Figure 2B; for an imagined single plasmid with an initial copy number of 10, we consider its representation, in terms of its presence in cells, after a number of periods, with conjugation both disabled (left-hand panel) and enabled (right-hand panel). If conjugation is disabled, then plasmids are essentially rapidly “flushed” from the system, as they are not transferred vertically. However, if we enable conjugation in our simulation, then plasmids are retained for longer within the system, suggesting that conjugation offers an important mechanism for stabilising a system long enough for switchable computations to occur. Although this considerably delays plasmid loss, the system will end up losing the repressed plasmid, since vertical transfer has a bigger impact than horizontal. This could be due to the 2D simulation setup, which prevents cells from having more frequent contacts (which is key for horizontal transfer), while vertical transfer is not space-dependent. Blocked plasmids, whose replication is repressed, do not pass through vertical transfer but do pass through horizontal (different replication mechanisms), which suggests that higher conjugation frequencies will delay plasmid loss for longer. For details on the simulation of conjugation, please see [26] and the Methods section of the current paper.

In Figure 2C, we show the results of spatially-explicit simulations of our system, in which plasmid A represses the replication of plasmid B. Both plasmids start off at roughly equal numbers, but we see that the red plasmid A rapidly dominates the population.

We then show (Figure 2D) how the system may be “switched”, such that an alternative computational task is selected for the population. Replication of plasmid A is repressed by an external signal, which leads to both a gradual loss of the red plasmid A, and an increase in the representation of the green plasmid B (since its replication is no longer being repressed by plasmid A). If we remove the external signal repressing plasmid A, then the system will gradually switch back to a dominant “red” state, and this process is indefinitely repeatable. In contrast to Figure 1B, this agent-based model was observed to be much more robust to rate changes. Using the same time-setup of Figure 2D, we increased the rate that governs the repression of replication of plasmid B (by plasmid A) tenfold—thus making such repression stronger. As a result, the number of plasmids B decreased rapidly at the beginning of the simulation, and the population took longer to recover when tasks where switched (see Appendix A). Nevertheless, despite time delays, the switching was successful, which suggests a high degree of robustness against rate variability.

The spatially-explicit nature of our simulation means that it is possible to analyse further the distribution of plasmids in the system. In Figure 3A, we show a snapshot of a simulation in which plasmid B (green) predominates. We see that the distribution of plasmid B is certainly not uniform, as can be observed in Figure 3B, where the cells with higher concentration are localized within the population. Almost all of these cells are touching each other and are clearly separated from the rest; the clusters are suggestive of both vertical and horizontal transfer. This clustering is responsible for stabilising the simulation, compared to the situation shown in Figure 1B (i.e., parameter values do not need to be unique). Clusters essentially act like plasmid reservoirs, and generate non-linear dynamics around the overall plasmid copy number, which favours robustness. In addition, these clusters add a different viewpoint to the analysis of bacterial differentiation within a population [27]—often a knowledge gap—which is advantageous to us. Indeed, a plasmid cluster (i.e., a local region with more copies of the repressed plasmid) favours the transfer of the weak plasmid over the strong, within itself. This region will then be where the task switching begins, since plasmids will be spread from clusters to the population. Clustering has a similar effect to natural bet-hedging strategies [28]. In these, cells within a population scout for beneficial fitness phenotypes in the case of an arrival of a sudden stress; if the stress comes, the population will recover from it thanks to the scouts (our clusters).

### 3.3. Distributed Computations Using Cell Consortia

An important recent development in synthetic biology and biocomputing has been the development of computational consortia; that is, computations that are distributed over a number of different cells, each of which performs a specific role [29,30,31]. This approach potentially allows for much more scalable cellular computation, as a large and potentially complex circuit may be broken down into smaller communicating components, each of which is placed in a specific cell. A common structure involves “sender” and “receiver” bacterial strains, each of which either transmits or acts upon specific signals, and we adopt that model here.

In Figure 4, we depict our scheme for multi-cellular computation of the Boolean NOR function, using four cell strains that interact to evaluate the gate. Recall that NOR is a negated OR function, so it returns “1” only when both of its inputs are zero, and “0” in all other cases (usefully, NOR is a universal gate, which means that any other Boolean function may be constructed using it).

Our system is composed of three plasmids; A (red) and B (green) represent the inputs to the NOR gate, and plasmid C (which defaults to blue) representing output = 1. Both A and B repress the output plasmid C, so C is only present (corresponding to an output value of “1”) if both A and B are absent (i.e., both input values are equal to zero). Each plasmid is represented by its own bacterial strain (the input plasmids in the “sender” strains), and we also use a fourth “computing” (or “receiver”) strain, which is engineered to express the appropriate fluorescent protein, according to the plasmid that it receives. The input/output strains are able to transfer plasmids horizontally to the computing strain, but cannot receive plasmids from others, which builds on the extensive toolkit for orthogonal conjugation systems [32]. The logic of the computation (Figure 4B) takes place at the “computing strain”.

We show the results of simulations for each of the four input cases (00, 01, 10, and 11): In the first case, we see only blue cells, as that is the only situation in which we expect to see an output value of 1. In the other cases, we see a preponderance of green (where the B input dominates), red (where the A input dominates), or a mixture (where both input plasmids are represented equally). This confirms the in-principle possibility of engineering plasmid copy numbers for the purposes of distributed cellular computation.

### 3.4. Use-Casing the Potential of Task Switching

We developed two simple models to demonstrate the potential of the suggested strategy (Figure 5). Two different use cases are shown: (1) Switch between two completely different tasks (Figure 5A), and (2) repurpose the meaning of the inputs to the same task (Figure 5B).

In the population simulated in Figure 5A, two plasmids coexist, each encoding an different inducible promoter with a fluorescent reporter downstream (red for plasmid A, green for plasmid B). These two tasks are different, in that they respond to different input signals (*s1* and *s2* respectively). Depending on which task runs at a given time, the cells will be sensing the corresponding signal. Therefore, this approach allows us to encode different tasks (e.g., biosensors) that will be active on demand. Figure 5A shows the performance of the simulation over 50 generations. Plasmids A are predominant until t=40, when they are externally repressed; as a result, plasmids B take their place. Simultaneously, the two input signals are changed over time—note that the values 0 and 1 indicate their absolute absence and their saturation, respectively. We see that, during the first 40 generations, only the dynamics of signal *s1* are captured by the population, while signal *s2* is ignored or captured at residual levels. The reverse situation occurs from t=40 onwards, when only signal *s2* is sensed.

The repurposing of the device (i.e., the same implementation, but different functions) is illustrated in Figure 5B, by changing the effects of the input on the circuit. Similar tasks as before coexist in a population, where both plasmids express a reporter. In this example, there is only one input signal, which is an inducer for the circuit in plasmid A and a repressor for that of plasmid B. By changing the meaning of the inputs, the computation returns a different output. The simulated population reads the signal as an inducer until t=40, when plasmid A is externally repressed. From then on, plasmids B will be present in higher copy numbers, and the same signal will be read as a repressor.

In all cases, task switching is heavily linked to memory management: The tasks that are not being used are sent back to memory. In a later step, those tasks can be recovered. This is the case of the interaction motif between plasmids (time = 40 in Figure 5), which suggests a potential approach to engineer multicellular memory circuits that could record the state of external signals.

## 4. Materials and Methods

### 4.1. Differential models

Ordinary Differential Equations (ODEs) were used to perform deterministic simulations of plasmid dynamics in the non-spatial model (Figure 1). The first set of ODEs describe the reactions $A+B\overset{k_{2}}{\to }A$ and $B\overset{k_{1}}{\to }2\cdot B$:(1)$$\frac{dA}{dt}=0$$
(2)$$\frac{dB}{dt}=k_{1}\cdot B-k_{2}\cdot A\cdot B,$$
where plasmid *A* is constant, $k_{1}$ = 0.05, $k_{2}$ = 0.005, and the initial conditions are *A* = *B* = 100 (all units dimensionless). Equilibrium is found at $k_{1}/k_{2}$ = 100 (Figure 1).

The circuits of Figure 5 were also simulated deterministically. Both models run inside cells of a spatial, discrete simulation. In spatial simulations, the dependence between the plasmids (control plasmid copy number) is being handled stochastically. These ODEs here refer to the circuits in the plasmids, whose function is independent from each other. The set of ODEs that govern the performance of the first circuit (Figure 5A) is:(3)$$\frac{dpA}{dt}=k_{2}\cdot pA^{a}-k_{1}\cdot s_{1}\cdot pA$$
(4)$$\frac{dpB}{dt}=k_{2}\cdot pB^{a}-k_{1}\cdot s_{2}\cdot pB$$
(5)$$\frac{ds_{1}}{dt}=k_{2}\cdot pA^{a}-k_{1}\cdot s_{1}\cdot pA$$
(6)$$\frac{dpA^{a}}{dt}=-k_{2}\cdot pA^{a}+k_{1}\cdot s_{1}\cdot pA$$
(7)$$\frac{dRFP}{dt}=k_{3}\cdot pA^{a}-k4\cdot RFP$$
(8)$$\frac{ds_{2}}{dt}=k_{2}\cdot pB^{a}-k_{1}\cdot s_{2}\cdot pB$$
(9)$$\frac{dpB^{a}}{dt}=-k_{2}\cdot pB^{a}+k_{1}\cdot s_{2}\cdot pB$$
(10)$$\frac{dGFP}{dt}=k_{3}\cdot pB^{a}-k4\cdot GFP,$$
where *pA* and *pB* are the promoter in plasmids A and B, respectively, $k_{1}$ = 1 is the rate of binding of the signals to their cognate promoters in either plasmid (promoters with signal bound are denoted by $pA^{a}$ or $pB^{a}$), $k_{2}$ = 50 rates the reversed reaction (unbinding) back to *pA* or *pB*, $k_{3}$ = 200 is the expression (merged transcription and translation) of the target gene in each plasmid, and $k_{4}$ = 1 is the degradation of the proteins GFP and RFP. The number of plasmids *A* and *B* is determined by the discrete simulation.

The circuit in Figure 5B was based on the same equations as above, but with the following changes: Signal *s2* is removed from the system, signal *s1* inhibits *B* and GFP is expressed from pB rather than $pB^{a}$. Therefore, Equations (Equation 5) and (Equation 10) change into:(11)$$\frac{ds_{1}}{dt}=k_{2}\cdot pA^{a}-k_{1}\cdot s_{1}\cdot pA-k_{1}^{r}\cdot s_{1}\cdot k_{2}\cdot pB^{a}$$
(12)$$\frac{dGFP}{dt}=k_{3}\cdot pB-k_{4}\cdot GFP,$$
where $k_{1}^{r}$ = 30, which is the rate of repression of pB by signal s1. All rates are expressed in molecules and hours, following values commonly used in mathematical models [33,34]. In any case, signals *s1* and *s2* are abundant or absent, therefore their derivatives can be considered null.

### 4.2. Stochastic Models

The conditional replication of plasmids is handled stochastically in the agent-based model. Gillespie’s algorithm [35] was used to calculate the intracellular performance of plasmid stability in Figure 2, Figure 3 and Figure 4. In Figure 2 and Figure 3, the reactions simulated were *A* + *B*
$\overset{k_{b}}{\to }$
*A* + $B^{b}$ and its reversal *A* + *B*
$\overset{k_{u}}{\leftarrow }$
*A* + $B^{b}$, where $k_{b}$ is the rate of plasmid *A* blocking the replication of *B*, and $k_{u}$ is the rate of unblocking such repression. Their values are 1 and 0.5, respectively.

Figure 4 includes a third plasmid, *C*, to perform a NOR logic function. The reactions for this simulation are: *A* + *C*
$\overset{k_{b}}{\to }$
*A* + $C^{b}$, *A* + *C*
$\overset{k_{u}}{\leftarrow }$
*A* + $C^{b}$, *B* + *C*
$\overset{k_{b}}{\to }$
*B* + $C^{b}$, and *B* + *C*
$\overset{k_{u}}{\leftarrow }$
*B* + $C^{b}$.

### 4.3. Agent-Based Spatial Simulations

For spatially-explicit simulations we used the agent-based tool DiSCUS [26]. This platform has previously been used to study the spread and growth of bacterial populations [36], and has been included in design-build-test synthetic biology life cycles [37]. In DiSCUS, a population of rod-shaped cells grows on a 2D surface. Each cell was coded to run a copy of either the stochastic or deterministic simulation under study. The spatial simulation resolved plasmid loss due to vertical transfer and plasmid gain due to horizontal transfer.

Vertical transfer takes place after cell division, when the mother transfers its plasmids to the two daughters. This is done randomly, so each plasmid has equal probability of ending up in each of the two new cells. For horizontal transfer (among siblings), two or three (random) plasmids are transferred in each conjugation event (copy number is not decreased in the sender) [38,39]. After each division and conjugation, plasmid copy numbers were updated, and the intracellular stochastic simulations adjusted the final numbers of repressed elements, accordingly. Importantly, vertical transfer will lose repressed plasmids (they do not replicate after cell division) but horizontal transfer won’t (they are copied in the recipient cell).

Time was measured in theoretical doubling-times, what we called generations (Figure 2), which is the time it takes for a rod-shaped body to grow and divide in DiSCUS, and corresponds to 450 iterations of the software. Conjugation frequencies where fitted to experimental observations (see [26] for details). Instead of pre-defining conjugation frequencies, DiSCUS implements the probability that a cell will conjugate with a neighbouring cell at any time-point (i.e., each of the 450 iterations) during its lifetime (probabilities range from 0.001 to 0.05). This parameter was then fitted to the frequencies obtained experimentally, both numerically in 3D solid cultures [40] and visually in 2D surfaces [41] (see [26] for comparison between DiSCUS and experiments).

## 5. Discussion and Conclusions

The engineering of increasingly complex tasks (i.e., genetic circuits) in cells is a major challenge, and a very active research topic. However, the design of management strategies for the execution of these tasks has received relatively little attention. Computer science, commonly used to frame the development of genetic circuits, has successfully achieved strategies to this end that can be of use to synthetic biology. Here, we present a task switching method in bacteria, as a way of managing cellular resources.

Task switching is designed by controlling the plasmid copy number (CN). Since each plasmid will encode for a specific task, the control of CN will result in the population running one task or another, without the need to re-design and re-engineer the cells. As envisioned here, this control is achieved via transcription factors [23]; that is, by making plasmid replication dependent on a repressor (for instance, using the LacI repressor to inhibit the replication of a replicon modified with the LacI operator (LacO) [23]). The efficiency of this repression can be controlled and fine-tuned, since not all regulators have the same affinity for their target promoter or noise patterns [42]. This implies that the strength of plasmid loss, through vertical transfer, can to some extent be engineered to adjust the computation. The more efficient this repression of replication is, the faster plasmids will be vertically lost. We use Horizontal Gene Transfer (HGT) as a tool to balance—and reverse—that loss. According to our simulations, HGT generates plasmid reservoirs that restore the equilibrium to an otherwise collapsing (i.e., inevitable plasmid loss) scenario. Another possibility to explore in future studies would be to engineer a second replicon, corresponding to low CN, on the plasmids. This way, the plasmids will never be completely lost through vertical transfer and HGT will allow for faster and more accurate task switching. Nevertheless, the potential control mechanisms over HGT [43] means that it lends itself to both single-strain and multicellular computations, based on this approach.

The implementation of analogue computations, in contrast to digital, is a potential benefit of this system. Initial and final states (i.e., plasmid A dominates over B/plasmid B dominates over A) are not discrete, but continuous (Figure 5C); even more, if the measurement focuses on a local region within the population. Moreover, unlike gene expression, which is a relatively fast event, and thus easily abstracted into Boolean values (regardless the level/threshold), the modulation of plasmid copy number is a rather slow process, where intermediate values are indeed functional. That makes copy number a promising programmable feature to exploit the almost unexplored field of analog computing in synthetic biology [44,45], in order to develop, for instance, memories or recorders [46].

Plasmids and HGT may play a major role in interbacterial relationships and the evolution of microbial communities [47]. Such powerful tools should not be left out of the synthetic biology toolbox. The ability to control the population of plasmids opens the door to engineer adjustable cell colonies that would switch to a different set of genes, depending on their needs. In the case of applications where cells must display optimal performance in changeable environments (e.g., bioremediation), task switching offers interesting solutions. This study demonstrates the in-principle feasibility of using them to achieve complex human-defined computations in cellular systems, and provides baseline information for their future wet-lab implementation.

## Figures and Tables

**Figure 1 life-09-00014-f001:**
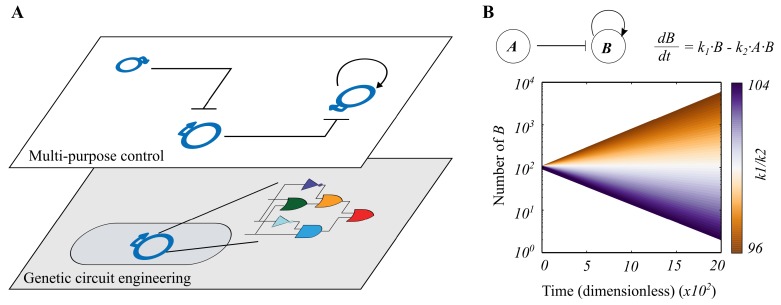
Overall design for a multi-purpose cellular computer. (**A**) The bottom control level encodes a single genetic circuit in a plasmid vector, which executes a single “program”. The top control level handles switching strategies for regulating the numbers of such plasmids in a cellular population. This is done via inhibiting plasmid replication or by promoting plasmid horizontal transfer (i.e., extra replication). (**B**) Deterministic analysis of a two-plasmid system (where plasmid A represses the replication of B, which, in turn, replicates via positive feedback) shows that the system is highly unstable; that is, plasmid B tends to either increase indefinitely or disappear.

**Figure 2 life-09-00014-f002:**
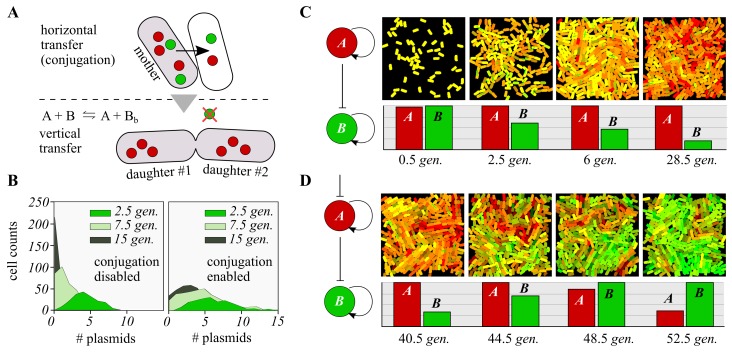
Switchable control of plasmid copy number through horizontal (conjugation) and vertical gene transfer. (**A**) A basic principle guides the following theoretical model: Plasmids whose replication is being repressed will not be spread through vertical transfer (i.e., mother to daughters), but can still be copied within siblings via horizontal transfer. (**B**) Distributions representing the number of plasmids in individual cells across a population. The rate of plasmid loss, when its replication is being repressed, is much faster than when conjugation is disabled (i.e., when plasmids are not transferred horizontally). This suggests that conjugation is a powerful tool for stabilizing the systems long enough to allow for switchable computations. **C**) Simulation of a population where all cells start with two plasmids, A and B. The overall number of B plasmids (bar plots) decreases over time, since its replication is repressed by A. (**D**) Same simulation as in (**C**), but the replication of plasmid A can be externally repressed. This repression over A happens after the overall number of plasmids B becomes very low (but not zero). As a result, the scenario is reversed and plasmid B predominates over A. Time in all plots is measured in generations (*gen*). Details on the simulation of conjugation is in the Methods section; y-axis on the histograms goes from 0 to 1, showing the relative difference between plasmids A and B during the computation, where plasmids with higher copy number is set to 1 and the other scaled accordingly.

**Figure 3 life-09-00014-f003:**
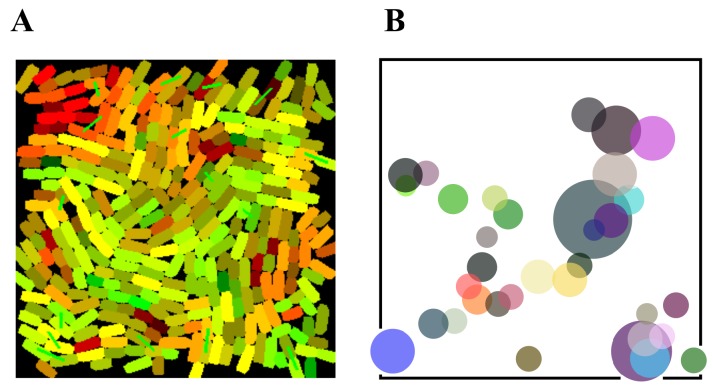
Plasmid copy numbers are not homogeneously distributed across a population, but highly clustered. (**A**) Snapshot of a simulation in which plasmids B (green) predominate. (**B**) Cluster formation. Dots in the plot show the positions of cells with more of plasmid B (green), in the simulation snapshot shown in (**A**). There is one dot per cell with high B concentration; the colour of dots is meaningless; diameter of dots are directly proportional to the plasmid copy number in each cell. Some dots are perfectly aligned, which suggest vertical transfer, while groups of cells (for example, in the bottom right) increased the copy number via conjugation.

**Figure 4 life-09-00014-f004:**
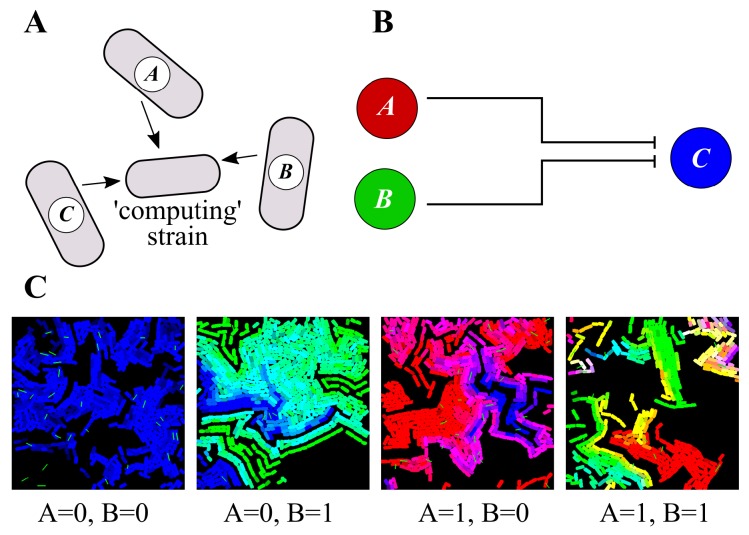
Multicellular computation in a 4-strain consortium. (**A**). There is one strain per plasmid, plus another computing strain—the input/output strains are able to transfer plasmids horizontally to the computing strain, but cannot receive plasmids from others. (**B**). Design of a 3-plasmid system that responds to a NOR logic function: Both input plasmids A and B repress the replication of output plasmid C. (**C**). Simulations of the four logic cases highlight the spatial localization of the computation. Only the computing strain is shown—black spaces in-between correspond to different “sender” strains.

**Figure 5 life-09-00014-f005:**
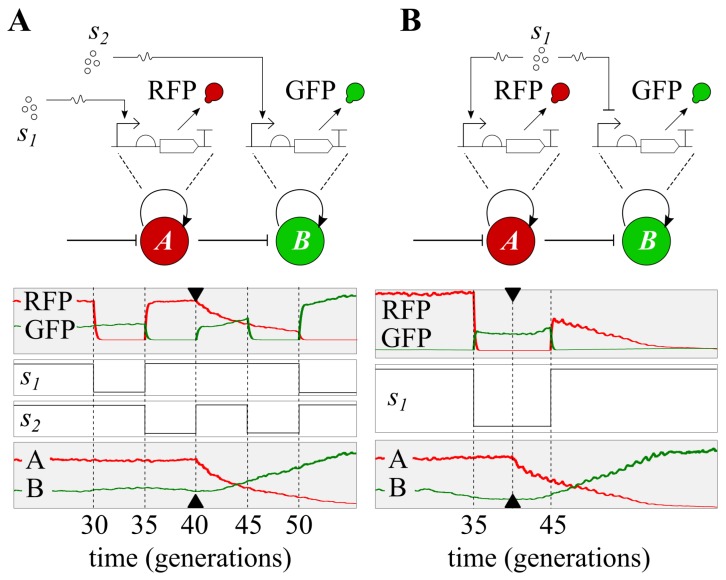
Use-casing the potential of multitasking. (**A**) Reduction of cellular workload. Two input signals, *s1* and *s2*, trigger the expression of different circuits—in this case, a simple reporter gene. By controlling the plasmid copy number, the population reacts to one of either two input signals. Cells will not have both circuits at the same time, thus reducing the metabolic cost. (**B**) Repurposing input signals. One input, *s1*, acts as an inducer for the circuit in plasmid A and a repressor for the circuit in plasmid B. Multitasking control allows for switching the population from using *s1* as an inducer to using *s2* as an inhibitor (and vice-versa). Arrows at time point 40 indicate when plasmids A are externally repressed (i.e., task switching takes place).

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
