# Peer review of "Dynamical Task Switching in Cellular Computers"

_life, 2019, doi:10.3390/life9010014_

Round 1

Reviewer 1 Report

The authors propose a model for a cellular population to contain multiple "programs" through maintaining multiple plasmids in a cellular population by controlling the plasmid copy number of different plasmids, having the inactive program maintained in a low copy in the population by suppressing its replication with the other plasmid and only allowing it's maintenance by conjugation, until an external signal inhibits the copy number of the first plasmid. This is an interesting mechanism and would be of use in synthetic circuit design if it is properly implemented.

However the authors need to be more specific with it's biological implementation, specifically I would like to see:

Examples of mechanisms for plasmid copy control mentioned in the introduction and their appropriateness for this type of system (the two plasmids need to have compatible origins of replication and both need to be able to survive in a high and low copy state for the system to function). 

An additional constraint is that one of the plasmids copy numbers must also be able to be controlled by an external signal.

A discussion on the rates of plasmid propagation via horizontal transfer versus vertical transfer, are the rates and frequencies biologically similar? If one of them is dominant, is this reflected in the model?

It would be helpful if there were at least a proposed biological implementation on the molecular level (genes and inducible molecules, i.e. tetR/lacI, aTc, IPTG) with a discussion of what parts have already been demonstrated or have precedent in the literature, in order to better visualize the system and know a way forward. LacI/LacO is mentioned in the discussion but has copy number control been demonstrated with this system? This could go in the introduction. There are groups like the Voigt lab that have used plasmid copy number control in synthetic circuits, so it might be worth looking at that and similar systems to propose an implementation.

On the computational side, the model has to be better described. At present it will be difficult to recapitulate the results of the model. In the list of ODEs for Figure 5, it seems like plasmids A and B are independent of each other and I can't see how plasmid A represses plasmid B, and how the signal will repress plasmid A, these need to be more explicitly defined. The way plasmids replicate vertically and horizontally also need to be described in the discussion on the model. The modeling code should also be made available with the publication.

Smaller notes:

In paragraph 5 of the introduction, Section 3 is said to describe experimental results but the paper is on modeling and simulations.

I think Equation (6) should be dpA^a/dt 

Author Response

PDF file attached.

Reviewer 2 Report

Please, see the .pdf attached.

Author Response

Review attached as a PDF file.

Reviewer 3 Report

The paper describes interesting computing system using population dynamics of plasmids in bacterial consortia.  In the proposed system, a plasmid replication in a cell can be modulated by itself or another plasmid.  After cell division, the system of the paper diminishes plasmids when their replication is inhibited.  Although detailed dynamics of plasmid replication is not described for spatial modeling, Fig 2A implies that each of two daughter cells has the same number of the plasmid with mother cell activating replication of the plasmid.  During conjugation, 2 or 3 plasmid molecules are “transferred” to a recipient cell.  Using this cell consortia system, the author proposed a distributed computation.  Additionally, the plasmid population dynamics directs switching of a task by the same inducer.

The paper does not have enough methods, results, and explanation of the system.  Without additional supplement with detailed descriptions, the paper cannot show acceptable conclusions.  Additionally, the paper did not show biological feasibility due to lack of specific biological mechanisms described in biological papers.  Without biological evidence, the paper should be categorized in information science field such as unconventional computing, rather than in synthetic biology.  Modeling in the paper does not have enough description to justify results from the modeling, too.

The mechanism of no vertical transmission is not clear.  If a plasmid is not degraded, one of the daughter cells has the plasmid. In other words, the plasmid is vertically transferred to the daughter cell.  Even if the model includes plasmid degradation, transfer speed of DNA must be higher than degradation speed.  The authors need to show such fast transfer mechanism.  From another view point, equation (2) at line 176-177 shows A-dependent degradation of B, rather than inhibition of replication.

In the case of F-plasmid of E. coli, the plasmid indeed has two replications origins, OriR and OriT.  However, the paper needs to show specific examples of biochemical mechanisms to allow horizontal transfer and not to allow vertical transfer.  Without such evidence in references, the proposed model is not feasible.

Even though the reviewer appreciates past publications with DiSCUS, the paper should show movies and parameters of the simulations. 

Physiological condition of cells and HGT rate would depends on cell density.  The simulation should include cell density effect described in a wet lab reference.

Line 211
After the transfer of the plasmid, is the copy number of the plasmid decreased, or kept? Although Fig 2B implies no decrease of copy number after the transfer, it should be clearly described.

Also, the paper should show wet-lab reference which shows multiple plasmid transfer during a single conjugation event.

Line 213
Considering the rod-shaped body, probability of conjugation rate may depend on an angle made by two cells next to each other.  When the two cells are close to each other, they can make "T", "=", "<" characters and so on.  The paper needs to refer results from wet-lab experiments.  Considering two daughter cells, in a sparse distribution of the cells, on agar plate and DiSCUS simulation tend to align along with their long axis, this possible angle dependency is crucial for the proposed system.

Fig 4
The paper needs to show the three orthogonal strain-specific HGT mechanisms from wet-lab papers.

Lines 150-171 and Fig 5
Although the reviewer understands equations (3)-(12), the reviewer cannot find how to draw lines for A and B in Fig 5.  Considering the limitations of the length of a paper in this journal, this section should be in another paper.

Minor point:
Vertical axis of Fig 1B should be log-scale.  Also, the line representing B=100 should be shown more enhanced than the other line. 

Author Response

Review attached as a PDF file.

Round 2

Reviewer 1 Report

The revisions are an improvement to the initial ms, and I believe my concerns have been addressed.